



# Mapping transition region flows to the ionosphere in a global hybrid-Vlasov simulation

Venla Koikkalainen[1], Maxime Grandin[2], Emilia Kilpua[1], Abiyot Workayehu[1], Ivan Zaitsev[1], Liisa Juusola[2], Shi Tao[1], Markku Alho[1], Lauri Pänkäläinen[1], Giulia Cozzani[3], Konstantinos Horaites[4], Jonas Suni[1], Yann Pfau-Kempf[5], Urs Ganse[1], and Minna Palmroth[1,2]

[1]University of Helsinki, Helsinki, Finland
[2]Finnish Meteorological Institute, Helsinki, Finland
[3]LPC2E, OSUC, University of Orléans, CNRS, CNES, Orléans, France
[4]Cooperative Institute for Research in Environmental Sciences, University of Colorado, Boulder, Boulder, CO, USA
[5]CSC – IT Center for Science, Espoo, Finland

**Correspondence:** Venla Koikkalainen (venla.koikkalainen@helsinki.fi)

**Abstract.** The dynamics of the inner magnetosphere and magnetotail are determined by a number of factors such as magnetic reconnection, plasma instabilities, and large-scale plasma motion. We use the global hybrid-Vlasov simulation Vlasiator to study these dynamics as well as their signatures in the ionosphere. We observe magnetic reconnection, fast flows, and vorticity in the transition region between the Earth's dipolar field and the magnetotail. In our simulation, reconnection is first triggered at

5  the dawn and dusk sides of the magnetotail current sheet. It then spreads across the current sheet. Concurrently, an azimuthally periodic, wave-like density structure develops in the transition region along with fast Earthward flows and enhanced vorticity patterns. The Earthward flows and vorticity induce field-aligned currents, which map onto the ionospheric simulation domain, creating a patchy current distribution. We find that the event is driven by the combination of reconnection-induced fast flows and the ballooning/interchange instability.

## 1 Introduction

The transition region between the Earth's dipolar magnetic field and the magnetotail is a highly dynamic region of space. In this domain, at distances of $\sim 6$ to $12\ R_E$ in the nightside magnetosphere, the Earth's dipolar magnetic field begins to stretch into the magnetotail, and phenomena occurring in both the inner and outer magnetosphere play a role in the stability of the region (Gabrielse et al., 2023). Additionally, the region is coupled to the Earth's ionosphere through magnetic field lines.

One of the prominent manifestations of the coupling between the magnetosphere and the ionosphere is the field-aligned current (FAC) system, i.e. electric currents carried by particle flows between the magnetosphere and the ionosphere. FACs are generally found to consist of two large-scale current pairs, shown e.g. in Fig. 1 in Carter et al. (2016) and Fig. 6 in Palmroth et al. (2021). The region 1 (R1) current is coupled to the magnetopause current, and the region 2 (R2) current is coupled to the partial ring current. From the ionospheric perspective, the R1 current is located at higher latitudes, and flows into the ionosphere on the dawnside, and out on the duskside, while the R2 current is located at lower latitudes and the flow directions are reversed

in comparison to R1 (Iijima and Potemra, 1978). In some studies, additional current loops flowing in the same direction as the





R2 current have been observed at latitudes higher than the R1 system. These have been termed as the R0 current, and they have been found to couple either to plasma flows in the magnetotail (e.g., Wang et al., 2022; Birn et al., 2020), or to the cusp during northward interplanetary magnetic field (IMF) conditions as reviewed by Milan et al. (2017).

Perhaps the most dramatic loss of equilibrium of the transition region occurs in the event of magnetic reconnection in the magnetotail. Tail-wide reconnection is associated with magnetospheric substorms (Angelopoulos et al., 1992); energetic events that launch plasma towards and away from the Earth. It is widely believed that magnetospheric substorms are driven by magnetotail reconnection which accelerate bulk flows of plasma away and toward the Earth and reconfigure the global magnetic field topology (Angelopoulos et al., 2008). The typical ionospheric signatures of a substorm include disturbances to

the Earth's magnetic field and bright and dynamical auroral displays, particularly at high latitudes (e.g., Akasofu, 1964; Sitnov et al., 2019). More localised reconnection is associated with bursty bulk flows (BBFs) that are often observed by spacecraft in the magnetotail at geocentric distances of 9–19 $R_E$ (Earth radii) in the nightside (Angelopoulos et al., 1992). These events are typically defined as having a bulk speed of at least $400 \, \mathrm{km \, s^{-1}}$, and are considered to be related to magnetic field dipolarization (e.g., Angelopoulos et al., 1992, 1994). Nakamura et al. (2004) found that the observed width of BBFs in the dawn-dusk

direction is on average 2–3 $R_E$ and 1.5–2 $R_E$ in the north-south direction.

     At the leading edge of BBFs, dipolarization fronts (DFs) are often created when the fast plasma flow causes magnetic flux pileup, changing the tail magnetic field to a more dipolar form (Runov et al., 2011). DFs are characterized by a sharp increase in $B_z$ (north-south) magnetic field component. After being accelerated in the tail, BBFs are slowed down by the Earth's dipolar field at the radial distance of ∼-10 $R_E$ (Dubyagin et al., 2011). As a consequence of the braking, there may also be rebound

flows back towards the tail (Panov et al., 2010). As the flows brake, they can induce vorticity (Birn et al., 2011) at the flanks of the flow channel.

     BBFs have also been found to create field-aligned currents in both observations (Nakamura et al., 2005) and simulations (Yu et al., 2017; Birn et al., 2019). The connection of BBFs to the substorm current wedge (SCW) has been studied by e.g. Birn et al. (2019) and Nishimura et al. (2020), who found that several fast flow regions can contribute to the FAC distribution. The

traditional view of the substorm current wedge (SCW) is of a single current loop that connects the tail current to the ionosphere (McPherron et al., 1973). However, it has been found that the current wedge may comprise of smaller "wedgelets" (e.g., Rostoker, 1998; Liu et al., 2015) of FACs related to multiple fast flow channels (BBFs) separated in the azimuthal direction.

     In the literature, there are several terms for similar phenomena in the magnetotail. Fast flows that appear similar to BBFs (with a width of several $R_E$ in the tail), are often termed as "bubbles" (Birn et al., 2004; Wang et al., 2022) with lower flux tube

entropy $S$ (defined in the next paragraph) than the surrounding plasma in the magnetotail. As the BBFs/bubbles travel closer to Earth, they may induce vorticity on the flanks of the fast-flow channels, i.e. the regions of Earthward flow. At smaller scales (with a width of hundreds to thousands of kilometres in the magnetotail), similar Earthward flows with depleted entropy $S$ are sometimes referred to as interchange "heads" (Panov and Pritchett, 2018; Panov et al., 2019).

     As mentioned, BBFs are often characterized as bubbles of depleted flux tube entropy $S$, which is defined as $S = PV^{5/3}$ (Birn

et al., 2009), where $P$ is plasma pressure and $V = \int dl/B$ is the flux tube volume. The flux tube volume is found by integrating along the magnetic field line ($B$). Bubbles with low entropy can result either from reconnection or the ballooning/interchange





instability (Pontius Jr. and Wolf, 1990; Chen and Wolf, 1993; Birn et al., 2004; Wolf et al., 2009; Birn et al., 2011). In their studies, Pontius Jr. and Wolf (1990) and Chen and Wolf (1993) show that current reduction at edges of the bubble with depleted density (and depleted flux tube entropy) leads to localized charge accumulation in the plasma sheet. As a result, the bubble

becomes polarized, and an electric field develops across it that drives $\mathbf{E} \times \mathbf{B}$-drift towards the Earth. The bubbles may be created by the ballooning/interchange instability, which is an analog to the well-known Rayleigh-Taylor fluid instability (Sharp, 1984). In the fluid instability case, a denser fluid on top of a less dense fluid creates a distinctive pattern as it sinks due to gravitational effects. For the ballooning/interchange instability, instead of gravity and density, the curvature of the magnetic field and the pressure of the plasma are the governing factors (Ohtani and Tamao, 1993). For the case of the Earth's magnetotail, signatures

of instability include e.g. wavy structure in plasma parameters, such as flux tube entropy and density (in the azimuthal direction) and possible interchange motion of magnetic field lines. During interchange motion, caused by changes in the flux tube entropy of each magnetic field line, field lines switch position relative to each other (for schematics and figures see e.g., Chen and Wolf, 1999; Lin et al., 2014).

There have been several different approaches to studying the ballooning instability in the near-Earth magnetotail. Liu (1997)

derived a dispersion relation for the instability in an ideal MHD case, expanding on the theory of Ohtani and Tamao (1993) where the authors suggest that the ballooning instability is related to the coupling of Alfvén waves and slow magnetosonic waves. Several studies after this, e.g. Ma et al. (2014), Ma and Hirose (2016), and Korovinskiy et al. (2019), agreed that the stability of the magnetotail against the ballooning instability is governed by plasma pressure and magnetic field curvature, but the assumptions made in deriving e.g. dispersion relations vary significantly between different authors. As the situation

is mathematically very complex due to the presence of magnetic field curvature, there is no clear consensus on a theoretical approach to the matter, as summarized by Rubtsov et al. (2018).

There have also been substantial modelling efforts to study the formation of ballooning/interchange instability. Within the MHD framework, e.g. Birn et al. (2011), and Wang et al. (2022) have modelled the ballooning instability related to low-entropy bubbles (similar to BBFs), often focusing on the flux tube entropy depletion $S$ as the main destabilization factor. In

the MHD studies, the instability is often studied through the dynamics of low-entropy bubbles in the magnetotail, where the scales are similar to BBFs and the bubbles may induce vorticity as they move towards the Earth. Guzdar et al. (2010) and Lu et al. (2013) focused on the relation between dipolarization fronts and the interchange instability, finding that the instability could be what causes events with multiple dipolarization fronts seen in the azimuthal direction. Sorathia et al. (2020) studied ballooning instability at smaller scales, also with an MHD simulation. They investigated flux tube entropy and the gradient

of the residual $B_z$ field, which is oppositely aligned to the entropy gradient, indicating possible instability. The resulting instability has scales of $\sim 4000$ km in the magnetosphere, significantly smaller than the previous MHD studies. In addition to the MHD approach, the instability has been studied with local Particle-in-cell (PIC) simulations by e.g. Pritchett and Coroniti (1997, 2010); Nakamura et al. (2016), and Panov and Pritchett (2018), where the scales are again often smaller, a few thousand kilometers in the magnetotail.

Signatures of the ballooning instability have been observed both in the ionosphere and in the magnetosphere. Panov et al. (2019) observed dipolarization fronts related to Earthward flows and azimuthally drifting interchange "heads" in the plasma



sheet, coupled to ionospheric current intensifications and auroral beading signatures. Xing et al. (2013, 2020) studied sub-storm events where MHD ballooning criteria were met, and corresponding auroral structures were observed in the ionosphere. Nishimura et al. (2022) studied auroral beading and determined the current directions for dawn/duskward drifting beads. Ohtani and Motoba (2023) also studied auroral beading, but did not find an immediate link between it and the ballooning/interchange instability, suggesting instead that auroral beading is caused by a process that is smaller in scale. Nakamura et al. (2001) observed dawn-to-dusk polarization in Earthward flow bubbles, in agreement with the theory by Chen and Wolf (1993). They found auroral streamers and pseudo-breakups to be related to 3–5 $R_\mathrm{E}$ wide regions of enhanced flow in the plasma sheet that could be approximated as bubbles. Wu et al. (2018) observed a wavy dipolarization front that could be caused by the inter-change instability and studied electron acceleration related to it. Despite the advances in the understanding of the processes in the transition region, the role of fast plasma flows in the region in combination with the ballooning instability is still not fully clear, with different simulation and observational approaches resulting in varying results.

In this study, we examine the appearance of large-scale flow channels, which induce vorticity in the magnetospheric transition region in the global hybrid-Vlasov simulation Vlasiator (Palmroth et al., 2018; Ganse et al., 2023). We observe characteristics of the ballooning/interchange instability as the vortex flows evolve into a symmetric structure over the nightside magnetotail at the radial distance of $X \sim -8R_\mathrm{E}$ GSE (Geocentric solar ecliptic). The flows emerge in the simulation after the inclusion of a new ionospheric boundary model (Ganse et al., 2025), and are also seen to map onto the ionospheric grid of the simulation, indicating the importance of magnetosphere-ionosphere coupling. The structure of the paper is as follows: the details of the Vlasiator code and the 3D simulation run used for this study are given in Section 2, followed by the results of the simulation and the analysis of how the vortices form and evolve in Section 3. After this, the results are compared to previous simulation and observation findings in Section 4, followed by the conclusions in Section 5.

## 2 Model

The magnetospheric model used in this study is the global hybrid-Vlasov simulation Vlasiator (Palmroth et al., 2018; Ganse et al., 2023). Figure 1 shows the simulation domain at 800 seconds in the run on which this study is based (see details be-low). Vlasiator uses the ion-kinetic hybrid approximation, solving the Vlasov equation for ions (protons in the case of the run discussed here) and treating electrons as a massless charge-neutralising fluid. The electromagnetic fields are obtained by solving Maxwell's equations, which are closed by the Ohm's law, including a Hall term and the electron pressure gradient. The computations for the ion population are done six dimensions, with three position space and three velocity space dimensions. Vlasiator has been used to study a wide variety of magnetospheric phenomena such as auroral proton precipitation (Grandin et al., 2023), the magnetospheric and ionospheric responses to a solar wind pressure pulse (Horaites et al., 2023), magnetotail reconnection and flux rope identification (Alho et al., 2023), and plasmoid release in the magnetotail (Palmroth et al., 2023). Computational advances to the model that enabled the 6D runs are detailed in Ganse et al. (2023).

The recent addition of an ionospheric boundary model to Vlasiator (Ganse et al., 2025) allows us to investigate magnetosphere-ionosphere interactions. With the new addition to the model, the magnetosphere is now electrostatically coupled to the iono-





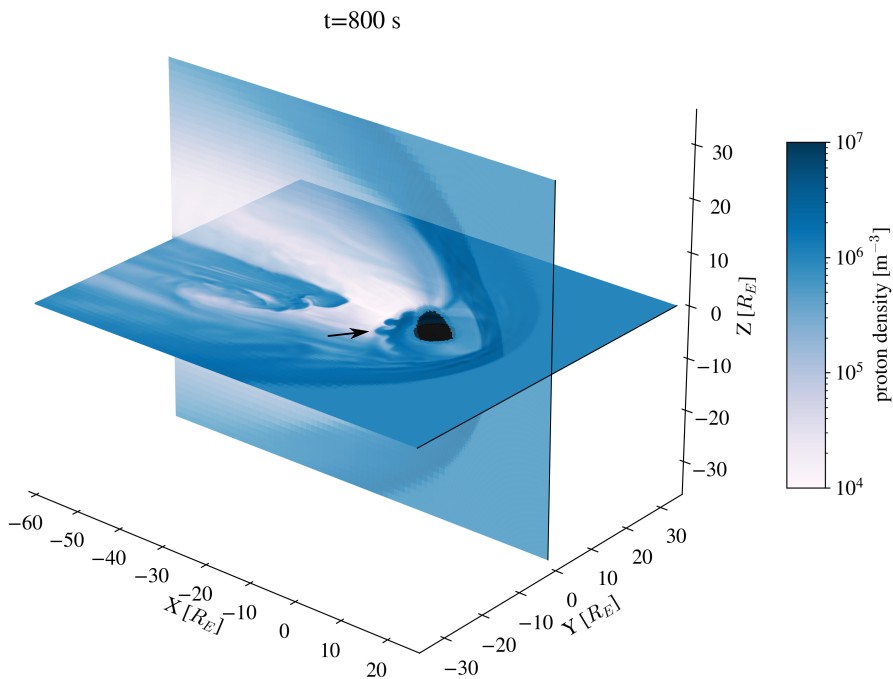

**Figure 1.** Part of the Vlasiator simulation domain, Y=0 and Z=0 planes, the axes are given in $R_\mathrm{E}$. The colours show the proton density in the solar wind and magnetosphere at t=800 s. The simulation used GSE coordinates, with positive X directed towards the Sun. The solar wind enters the simulation box from the +X boundary, and flows in the -X direction. The vortex structure studied in this paper is indicated with an arrow on the nightside.

sphere with a strategy analogous to MHD simulations (e.g., Janhunen et al., 2012; Goodman, 1995). In the new approach, simulation conditions are fed from the magnetospheric domain to the spherical ionospheric grid through magnetic field lines, using a dipole field within the inner boundary of the simulation. The ionospheric boundary model is height-integrated, with all parameters integrated from ground level to an altitude of 200 km. The input parameters for the ionospheric model are FAC density $j_\parallel$, electron density $n_e$ and temperature $T_e$. The details of the ionospheric boundary model are given in Ganse et al. (2025).

In the magnetosphere, the computational costs of running the simulation are alleviated by the use of adaptive mesh refinement (AMR). In the run used for this study, the spatial resolution was 8000 kilometers at the lowest, with three steps of refinement applied to certain parts of the grid, resulting in 1000 km as the highest spatial resolution. The highest resolution is applied to the magnetotail current sheet. The simulation uses the GSE coordinate system. In the simulation there is no tilt to the Earth's dipole magnetic field, and so the dipole axis is aligned with the Z-axis. The solar wind conditions for the simulation are given by the constant boundary conditions at the +X side of the simulation box. For the run discussed in this study, the solar wind speed was set to be constant at $750\ \mathrm{km\,s^{-1}}$ towards the -X direction. The interplanetary magnetic field (IMF) was defined to





be purely in the -Z direction with a magnitude of 5 nT. The density of the solar wind was set at 1 cm$^{-3}$ and the temperature at 0.5 MK. The full duration of the run was 1612 seconds, with output generated at 1 second intervals. The size of the simulation box is from -110 to 50 $R_\mathrm{E}$ in the X-direction and from -58 to 58 $R_\mathrm{E}$ in the Y and Z directions. In this run protons are the only ion species.

## 3 Results

### 3.1 Overview of flow channel formation

The event begins with magnetic reconnection starting on the duskward (Y>0) transition region of the magnetotail at X$\sim$ $-10R_E$ and Y$\sim 7R_E$. As the simulation progresses, reconnection is initiated on the dawn side (Y<0) of the magnetotail as well. This can be seen in Figure 2, which shows zoomed in sections of the Z=0 plane in the nightside tail. This plane is close to the magnetotail current sheet. Due to flapping motion of the current sheet, the Z=0 plane may not exactly coincide with the current sheet, but we can study relevant dynamics by looking at this plane. Figure 2 shows proton density, proton velocity $v_x$, magnetic field $B_z$, electric field $E_y$, pressure, and FACs at four time steps during the simulation. The FACs are shown as a function of magnetic local time (MLT) and geomagnetic latitude on the ionospheric grid of the simulation. The grey/pink contours on the magnetospheric figures mark $\pm 400$ km s$^{-1}$ ion velocities. Supplement S1 shows an animation of the proton density, $v_x$ and the FACs between t=580 s and t=880 s.

Figures 2a and 2b show a decrease in density, first on the dusk side, and then on the dawn side. At the same time, panels 2e and 2f show a region of strong Earthward flow, coinciding with the start of reconnection. The peak values of $v_x$ are approximately 600 km s$^{-1}$, clearly exceeding the BBF threshold 400 km s$^{-1}$, and they coincide with the regions where density is lower. The reconnection starting on the western/dusk side of the magnetotail is possibly due to Hall effects causing favourable conditions for reconnection, as has been observed in previous hybrid simulation studies (Lin et al., 2014; Lu et al., 2016). The resolution of the simulation is also lower at the flanks of the magnetotail compared to the rest of the magnetotail, which likely plays a role in the initiation of reconnection, as will be discussed further in Section 4. As the magnetotail current sheet thins, ions become decoupled from the magnetic field. At the same time, a Hall electric field (not shown in the figure) forms in the Z-direction (towards the neutral plane, $B_x = 0$), which causes plasma to drift towards dawn. This results in a thinning of the current sheet on the dusk side, making it more prone to reconnection (Lu et al., 2016).

The formation of the flow channels is closely related to the spreading of the reconnection across the magnetotail. The X-line can be seen in Figure 2f at t=680 s as a flow reversal of Earthward/tailward flow forms at about X=-12 $R_E$. Density undulations and narrow Earthward flow channels form in the transition region, as seen in panels 2c – d (density) and 2g –h ($v_x$). As the flow propagates closer to Earth, rebound flow in the tailward direction ($v_x < 0$) is created. Additionally, the flow braking induces vorticity, which will be discussed more in Section 3.3.

At the last time step, t=880 s, shown in Figure 2d the density structure has dissolved into a mushroom-shaped pattern, which grows in scale but diminishes in density. In the corresponding $v_x$ plot (panel 2h) we observe that rebound flows on smaller scales (compared to panel 2g) begin to form.





**Figure 2.** The evolution of the flows in the magnetosphere Z=0 plane and the ionosphere for four times in the simulation, with the columns each showing one time. Panels (a-d): proton density. Panels (e-h): ion $v_x$. Panels (i-l) $B_z$, Panels (m-p) $E_y$. Panels (q-t) pressure. The contour lines indicate 400 km s$^{-1}$ (grey) and -400 km s$^{-1}$ (pink) velocities. Panels (u-x): FACs on the ionospheric grid of the simulation, as a function of MLT and geomagnetic latitude, with red signifying current flowing into of the ionosphere, and blue meaning current out of the ionosphere.





Panels 2i–l show that the Earth's dipole field dominates the magnetic field $z$-component. At t=780 s (Fig. 2k), we see that the magnetic field $B_z$ is enhanced in the regions where the fast plasma flows (grey curves) are decelerated in the transition region. The $B_z$ enhancement is linked to dipolarization, a process influenced by the transport of magnetic flux through Earthward flows produced during reconnection. Panels 2m–p in Figure 2 show $E_y$, which is positive for regions of Earthward flow, corresponding with the idea of polarization leading to enhanced electric fields (Pontius Jr. and Wolf, 1990). The electric field pattern, with positive and negative $E_y$, is similar to the $v_x$ structure in terms of Earthward flow being associated with positive $E_y$, and vice versa. This suggests that the convective electric field is a major part of the total electric field. Panels 2q–t in Figure 2 show that pressure is increased on the Earthward side of the fast flows. In panels 2s and 2t it can be seen that the contours of the fast flow match the structure of the increased pressure. At the last time step (panel 2t) the pressure has started to decrease considerably.

At t=580 s, it can be seen in panel 2u that the FAC structure (with red signifying current flowing into the ionosphere, and blue signifying current flowing out of the ionosphere) generally resembles the traditional picture of the R1 and R2 currents. At this time, in the dusk side an additional along with an additional upward current is created between 20 and 22 MLT. At t=680 s in panel 2v the current patterns remain similar to the start of the event. In the last two panels (2w-x) large-scale patches of FAC structures form between 20 and 04 MLT. It can be seen that the original clear R1 and R2 current pattern starts to shift into a patchy distribution, where the boundary between downward and upward currents becomes wavy. The current flowing into the ionosphere intensifies, going from a maximum of $\sim 0.5 \mu\mathrm{Am}^{-2}$ at t=580 s to a maximum of $\sim 0.7 \mu\mathrm{Am}^{-2}$ at t=780 s. After this the inflowing current starts to become weaker again. The maximum values of outflowing current do not follow the same trend, but are higher at the start and end of the event ($\sim 0.76 \mu\mathrm{Am}^{-2}$ at t=580 s, $\sim 0.84 \mu\mathrm{Am}^{-2}$ at t=880 s) and lower for the the times in the middle ($\sim 0.67 \mu\mathrm{Am}^{-2}$ at t=680 s, $\sim 0.69 \mu\mathrm{Am}^{-2}$ at t=780 s). Between t=780 s (panel 2w) and t=880 s (panel 2x) the areas of the FAC patches grow. The change in FAC structures is seen predominantly at lower latitudes on the equatorward edge of the current structure. The connection between the FACs and the fast flows is explored in more detail in Section 3.3

As for the scales of the flow, in the magnetosphere the wavelength (defined as the distance between the plasma density maxima at a radial distance of $\sim 8$ $R_\mathrm{E}$) of the vortices is about 3.5 $R_E$ prior to the "mushroom"-like dissipation. In the ionosphere, the wavelength of the FAC structures is about 2000 km. The way the density variations evolve closely resembles the behaviour expected from the Rayleigh-Taylor instability and its plasma analogue, the ballooning/interchange instability, which will be discussed in more detail in Section 3.2.

Figure 3 shows the flow structures in the YZ-plane at X=-8 $R_E$ at t=790 s, when they are at their clearest before they start decaying. At this distance in the tail, the size of the structures is about 3 $R_E$ in the Z-direction. Positive $v_x$ (panel 3a) and positive $E_y$ (panel 3b) correlate with each other within the structures. In the magnetic field Z-component (panel 3c) we see an increase in the regions where the bulk flow is directed towards the Earth ($v_x > 0$) and a decrease when the flow is directed towards the tail ($v_x < 0$). This implies a buildup of magnetic flux at the leading edge Earthward flows.





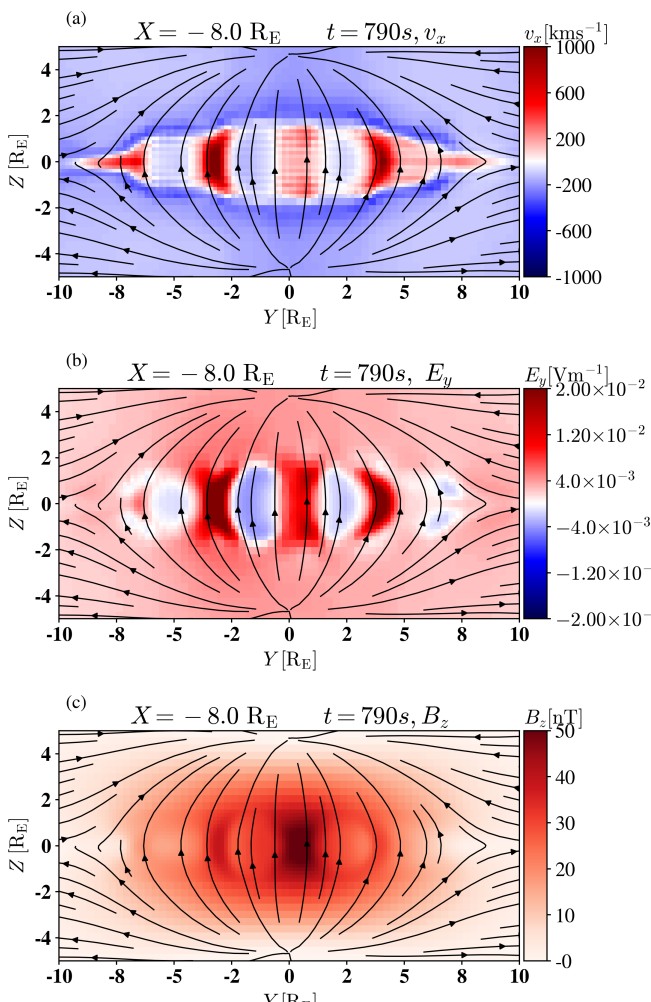

**Figure 3.** The transition region structures from the nightside magnetotail at X=-8 $R_E$ at $t = 790$ s. The black curves show the magnetic field lines in the plane. Panel 3a shows $v_x$, panel 3b $E_y$, and panel 3c $B_z$.



## 3.2 Signatures of the ballooning/interchange instability

As mentioned in Section 3.1, the evolution of the density gradient at the transition region resembles closely the fluid Rayleigh-Taylor instability, whose equivalent in a curved magnetic field plasma is the ballooning/interchange instability. The stability of the magnetotail against the ballooning/interchange instability can be studied through analysis of $B_z$ and entropy $S$ gradients, and changes in pressure $P$, as was done by e.g. Sorathia et al. (2020) and Birn et al. (2009). The ballooning/interchange instability is governed by plasma pressure and the curvature of the magnetic field. Looking for changes in these parameters can indicate changes in the stability of the region. In the magnetic field, we can look for stretching of the current sheet, which increases the curvature and thus changes the conditions of the magnetotail, possibly making it unstable. Birn et al. (2009) wrote that the magnetotail is stable when pressure decreases monotonically towards the tail, while entropy increases. Entropy may decrease in the tail due to loss in either flux tube volume or pressure. One example of entropy loss is reconnection pinching off part of a flux tube, thus decreasing its volume. Particle precipitation may also play a role in entropy loss, but it is not expected to be significant compared to magnetospheric sources of loss (Wolf et al., 2009). It is expected that variation in entropy with radial distance in the tail direction may cause loss of equilibrium.

Figure 4 shows the Z-component of the residual magnetic field $\Delta B$, defined here as the difference between the initial magnetic field (dipole field and constant IMF) at the start of the simulation, and the magnetic field at a particular time step. Additionally we plot the flux tube entropy S for the same times as in Figure 2. This analysis is similar to that done by (Sorathia et al., 2020). The flux tube entropy is calculated as an integral over closed magnetic field lines, which is why parts of the figure are empty (white), in the regions where the field lines are open. At the start of the event, most of the transition region exhibits a tailward gradient in $B_z$, consistent with the stretched field lines prior to the onset of reconnection. At the Earthward flows, $B_z$ can be seen to increase from negative values towards zero, as seen in panel 4a on the dusk side. As the simulation progresses, there is a similar increase in $B_z$ along the Earthward edge of the fast flow region, especially clear in panels 4c and 4d.

Figures 4a and 4e show that the entropy is decreased in the regions where there is Earthward flow and an increase in residual $B_z$, such as the fast flow region on the dusk side. Comparing to Figure 2, these are also the regions of low density. In regions of high pressure (Fig. 2), the flux tube entropy $S$ is also high, as is expected by the definition of $S$. At time steps $t = 780$ s and $t = 880$ s (panels 4g and 4h) there is a clear drop in entropy that coincides with the regions of fast flow. The wavy structure that forms from high values of entropy (most clearly in panel 4g) mirrors the regions of high pressure and high density.

As mentioned, the non-uniform growth of flux tube entropy towards the tail is an indicator of the ballooning/interchange instability (e.g., Birn et al., 2009; Wolf et al., 2009). For a stable situation the flux tube entropy would be expected to uniformly increase with distance from Earth towards the tail. The reduction of entropy can be attributed to reconnection and the loss in density and flux tube volume that is related to the plasmoid released in the tailward direction. This could also be the case in our simulation, where we see a decrease in entropy in regions where reconnection is triggered and density decreases. The fact that we see a decrease in entropy corresponding to the first Earthward flow that starts the event implies that the flow region is ballooning unstable (Birn et al., 2009).





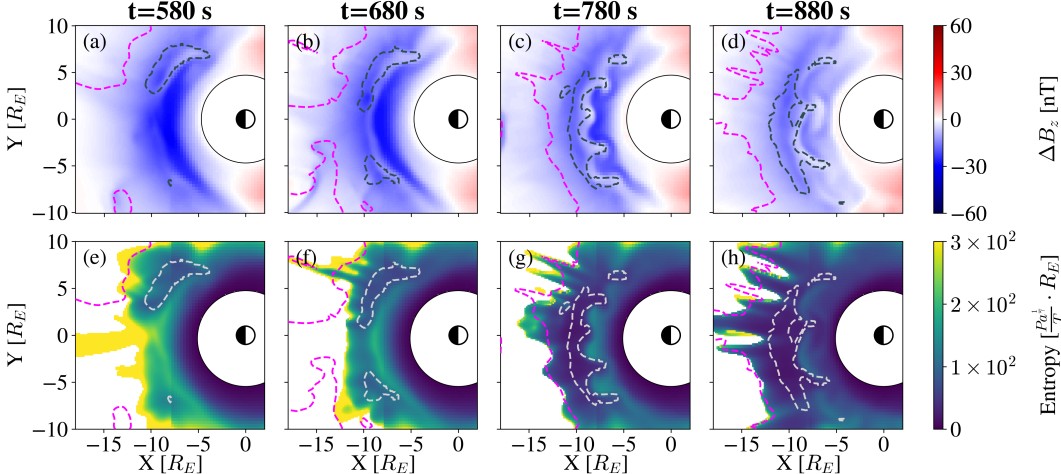

**Figure 4.** Panels (a-d): Residual $B_z$, i.e. the magnetic field minus the Earth's dipole field ($\Delta B_z$). Panels (e-h) flux tube entropy $S$. All panels are in the Z=0 plane, with $\pm 400 \mathrm{km\,s}^{-1}$ velocities marked with grey/pink contours. The four time steps are the same as in Figure 2

The ballooning instability is thought to induce density variations such as the ones in Figure 2, and the motion and stretching of magnetic field lines (e.g., Lin et al., 2014; Birn et al., 2015; Sorathia et al., 2020). To study this, we follow the motion of selected field lines. We choose 18 field lines (initially three rows of six field lines close to the region where the first Earthward flow begins), and track the motion of their magnetospheric seed points as the simulation progresses. This is shown in Figure 5, where field line placement is shown for six times in the simulation. The colouring of the lines has no physical significance and is used simply to track the motion of the lines in relation to each other. The field line motion is shown against the magnetotail current sheet, where we plot the X-component of velocity. At t=555 s the field lines are still grouped close together, and as the simulation progresses they start to drift apart and twist around each other. The twisting can be seen for the three last time steps, where we see interchange motion of the field lines relative to each other. This is highlighted for three field lines, marked with A (light yellow), B (cyan) and C (fuchsia). At t=705 the field lines are, from left to right, in the order A,B,C. By t=805 s we see that the ordering has changed to C,B,A, i.e. field lines A and C have changed positions. In addition to the twisting and interchange, there is considerable azimuthal drift of the field lines.

In summary, we found that there is clear variation in flux tube entropy in the magnetotail, in the X and Y directions, coinciding with regions of fast flow. Additionally we found interchange motion of field lines. These findings indicate that the structures in e.g. velocity and density are due to the ballooning-interchange instability.







**Figure 5.** A zoomed in section of the magnetotail, where we trace selected field lines on the current sheet plane as the simulation progresses. The colours are used to distinguish the field lines from each other. The colormap shows the x-component of velocity in the magnetotail current sheet (where $B_x$=0). The axes are in $R_E$. Three field lines are marked with A,B and C to highlight the switched positions from 705 s to 805 s.





### 3.3 Ionosphere-magnetosphere coupling and dynamics of fast flows

Finally, to investigate the effects of the fast flows, along with coupling between the magnetospheric and ionospheric domains, we plot the ion velocity $v_x$ and the vorticity on the magnetotail current sheet $(\nabla \times \mathbf{V})_z$ and study how they connect to the FACs in the ionosphere.

The coloured regions in Figure 6a show $v_x$, while plasma motion in the current sheet is indicated with the black arrows. In Figure 6b the colours indicate the z-component of vorticity. To find where the field lines from points of high vorticity map to, we plot the ionospheric grid in Figure 6c. Comparing figures 6a and b, we see that the regions of high vorticity are located at the flanks of the Earthward flow channels. Thus it would appear that the Earthward flow induces vorticity in the transition region. Following the field lines, counter-clockwise (positive vorticity) plasma motion in the current sheet corresponds to FACs flowing away from the ionosphere (blue colour in the figure), and vice versa. The FAC flow that is associated with each Earthward flow channel is oriented in the R1 sense in terms of direction of the current, as is to be expected based on the literature (e.g. Birn et al., 2004; Yu et al., 2017). The change to FACs is seen mostly in the R2 region, meaning the pair of FACs at lower latitudes.

From these figures we see that in addition to the ballooning/interchange instability discussed in the previous section, the dynamics of the fast flows also play a key role, at least in creating the FAC structure seen in the ionosphere.



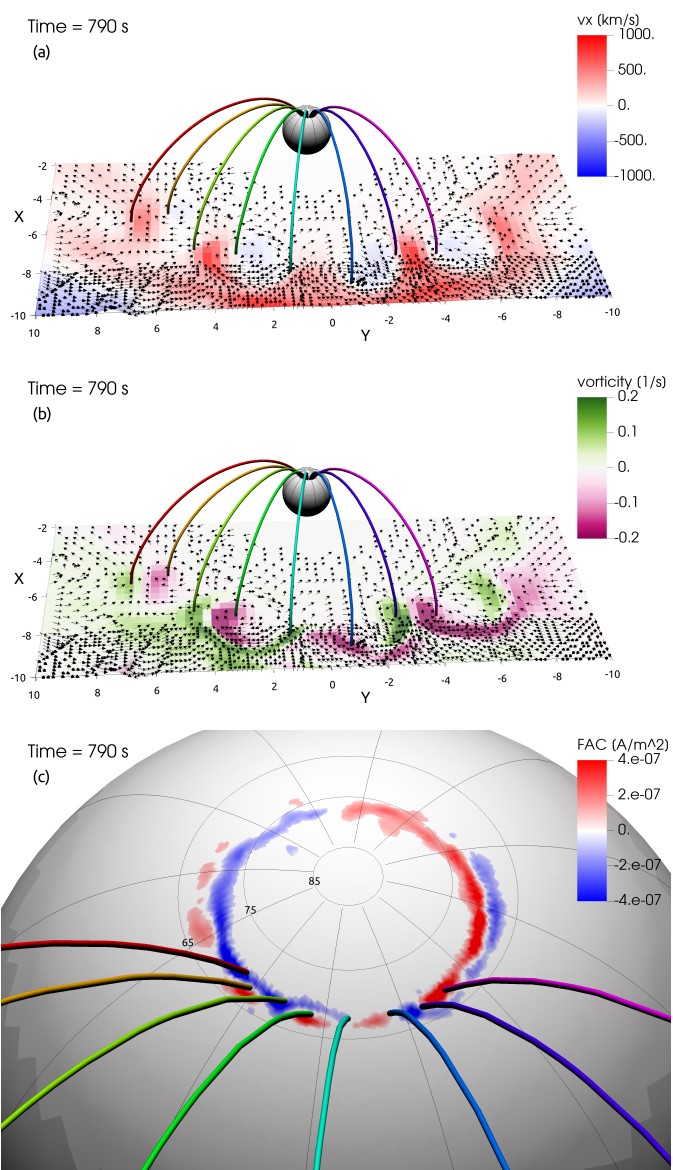

**Figure 6.** The coupling of magnetic field lines to the ionospheric grid. Panel (a) shows the velocity $x$-component in the current sheet ($B_x = 0$). The arrows point to the direction of the plasma velocity in the current sheet. The colouring of the traced field lines has no physical meaning. The axes units are in $R_E$. Panel (b) shows via the colormap the vorticity of bulk flow $(\nabla \times \mathbf{V})_z$ on the current sheet. Panel (c) is a zoomed-in section of the ionospheric grid, where the same field lines as in panels (a) and (b) can be seen to map to the ionosphere. Here the colormap shows FAC strength.



## 4 Discussion

In our simulation we have observed the creation and evolution of structured fast Earthward flows that create vorticity and FACs coupling to the ionosphere. The wavelength of the vortices in the transition region is about 3.5 $R_E$, and 2000 kilometres in the ionosphere. In previous literature, similar Earthward flows and the ballooning/interchange instability have been studied in

a variety of simulation and observational studies. We summarize below key previous studies to place our findings in context.

The event, i.e. the creation of vortex flow, seen in our ion-kinetic simulation most closely matches the spatial scales seen in MHD simulations of low-entropy Earthward flows, such as the ones reported by Birn et al. (2011); Birn and Hesse (2013); Birn et al. (2015). Birn and Hesse (2013) studied the substorm current wedge in relation to Earthward-propagating low-entropy bubbles. They found Earthward flows that break into narrow channels, which resulted in azimuthally spread ionospheric signatures,

similar to our study. They speculated that this effect is a combination of a cross-tail mode such as the ballooning/interchange, in combination with reconnection spreading over the night side. This scenario is supported by our simulation.

Guzdar et al. (2010) and Lu et al. (2013) studied the interchange instability in 2D MHD aiming to model the creation of multiple dipolarization fronts. They used a seed perturbation of the BBF scale (1-3 $R_E$) to initiate the instability, and found that the scale of this initial perturbation controlled the scale of the resulting wavelength of the instability. Lapenta and Bettarini

(2011) used a 3D MHD simulation to find self-consistent seeding of interchange instability in dipolarization fronts, following magnetic reconnection which was related to the kink instability. The scale of the kink instability determined the scale of the structures formed by the interchange instability. The scales seen in our study are similar to the BBF scales in Guzdar et al. (2010) and Lu et al. (2013). The results from these studies resemble our case, where the wavelength of the azimuthal structure is similar to the initial flow region that forms at the dusk flank of the magnetotail as result of reconnection. The multiple

dipolarization fronts are similar to the azimuthal structure we observe. In terms of observations, there have been THEMIS satellite observations of a wavy dipolarization front, studied by Wu et al. (2018).

More recently, Sorathia et al. (2020) studied the ballooning/interchange instability in a substorm growth phase, using a 3D MHD simulation coupled to an ionospheric solver with constant Pedersen conductance and no Hall conductance. The ballooning/interchange scales seen in their study (4000 km in the magnetosphere) were much smaller than in previous MHD

studies, and are closer to the ones found in PIC studies. Conversely to the previously mentioned MHD studies, the initiation of the instability in this study did not seem to be related to reconnection, but arose from a tailward $B_z$ gradient and a decrease in flux tube entropy, which are considered as possibly unstable to the instability. The magnetospheric structures were mapped onto the ionospheric grid of the simulation, where the scales matched that of auroral beading. While the conditions for instability (entropy and $B_z$ gradients) are similar in our simulation, the key difference that possibly explains the difference in scales is

that our simulated event is driven by fast flows that result from reconnection, while their simulation focuses on the substorm growth phase prior to reconnection onset.

The PIC approach has also been widely applied to the study of the ballooning/interchange instability. In the earlier studies, by e.g. Pritchett and Coroniti (1997) and Nakamura et al. (2002) the scales are on the order of Earth radii, similarly to our simulation. In later studies the scales observed in PIC have been smaller, with the kinetic effects starting to affect the size of



the structures (Pritchett and Coroniti, 2010, 2011, 2013). While Vlasiator can also be used to study the kinetic physics of the magnetotail, in the current simulation the large scale flow regions appear to dominate the picture, and thus the result more closely resembles previous MHD studies.

    Xing et al. (2013) used a conjunction of spacecraft and THEMIS all sky imagers, along with imagers to study the auroral response to the ballooning/interchange instability. They found evidence of the instability initiated at X=-11 $R_E$ in the magneto-

tail, with a wavelength of 1-3 $R_E$. The auroral response to this was the creation of additional wavelike structures on preexisting auroral arcs. These scales are similar to what we see in our simulation. Xing et al. (2020) continued this work and suggested that the ballooning/interchange instability resulted in auroral wave structures, but did not necessarily lead to substorm onsets.

    The auroral response to the ballooning/interchange instability is often thought to be auroral beading, i.e. an auroral form consisting of azimuthally separated auroral patches (e.g., Motoba et al., 2012). The ionospheric scales of these auroral bead-

ing events are smaller than what we observe in our simulation. Nishimura et al. (2022) used THEMIS all sky imagers and satellites, and found that the beading events studied matched the scales of e.g. the simulations by Sorathia et al. (2020). The magnetospheric data from THEMIS indicated that ballooning/interchange was a possible candidate for the creation of the auroral beading. Conversely, Ohtani and Motoba (2023) found that auroral beading should be attributed to a process on a smaller scale than the ballooning/interchange instability, and that mesoscale or large scale convection could not control auroral beading.

As is evident from the summary of previous research, the conditions and features of the ballooning/interchange instability and the resulting ionospheric and magnetospheric signatures vary between the different simulation/observation approaches. The event observed in our simulation shares features similar to many previous studies, but also presents distinct differences. The scales we observed are similar to many of the MHD studies which have reconnection as the main driver of the instability.

    Magnetic reconnection appears to play a key role in the development of the instability and determining its observed scale. In

our simulation, reconnection is initiated at the dawn and dusk flanks and then spreads across the magnetotail. This behaviour could result from the flanks having a coarser simulation grid and consequently relatively higher numerical diffusion compared to the mid-tail region. The reconnection creates an azimuthally symmetric, wave-like structure, evident e.g. in density (Fig. 2c). As the simulation progresses, this wave-like structure extends from the lower resolution to the higher with the same wavelength in both regions. The initiation of the reconnection on the dusk side of the magnetotail agrees with previous hybrid simulations

(Lin et al., 2014; Lu et al., 2016), where Hall effects were causing favourable conditions for reconnection primarily at the dusk. A Hall field is induced also in our simulation, suggesting that the combination of the resolution and the physical effects sets the conditions for the onset of reconnection. The creation of the vortex flows occurs at an early stage in the simulation run, where the magnetosphere is still in its phase of global reconfiguration caused by tail reconnection. Compared to in situ observations, this stage could resemble a case with a very elongated tail.

The balance between fast flows creating vorticity and the ballooning/interchange creating a similar structure is an interesting matter. While the Earthward flow is arguably an important driver of the observed structures, causing rebound flows and vorticity, our results suggest that the ballooning/interchange instability also plays a role. We observe a decrease in entropy with increasing radial distance from Earth (Figure 4), indicative of instability to the ballooning/interchange. There is an initial decrease in entropy where the reconnection first sets in at dusk/dawn. Additionally, later in the simulation we observe the interchange





motion of the field lines (Figure 5). The emergence of "mushroom-like" density structures (Fig. 2 panel (d)) is also an indicator of an instability, visually resembling the Rayleigh-Taylor fluid instability. It appears that the Earthward flow seeds the instability similarly to the studies by Guzdar et al. (2010), Lu et al. (2013), and Lapenta and Bettarini (2011).

Another factor to be considered is the effect of the ionospheric boundary model on the creation of the vortex flows. The current Vlasiator ionospheric boundary allows field lines to move in the ionosphere. This enables the observed interchange motion, and allows for the creation of vorticity. Studying the effects of the ionospheric boundary in more detail could be beneficial: For instance, it would let us see how the conductivities affect global magnetospheric convection. It would seem that in this case the reconnection in the magnetotail is the main driver of the instability rather than e.g. the ionospheric conductivity model. This is because the creation of the vortex flow coincides with the initial sites of reconnection in the magnetosphere.

The observed scales we see in the magnetosphere match those of BBFs, similar to e.g. Birn et al. (2011). This finding is in agreement with the event being primarily driven by magnetic reconnection. Figure 6 shows the mapping of field lines from the current sheet onto the ionospheric grid, where it can be seen that the FAC pattern is created in the typical R1 sense that is associated with BBF-like flow. In the ionosphere, the scales perhaps most closely match those of substorm wedgelets, similarly to Nishimura et al. (2020) who surveyed substorms with wedgelet type current loops, and found the average size of the wedgelets was found to be $\sim 3.2\,R_\mathrm{E}$ in the azimuthal direction in the magnetosphere and $\sim 600\,\mathrm{km}$ in the ionosphere.

## 5 Conclusions

In this paper we study the appearance of large-scale vortex flow in the transition region between the dipole field and the magnetotail in a global hybrid-Vlasov simulation. The main finding of the paper is that this appearance of large-scale vorticity inducing flows is triggered by magnetic reconnection, which then enables the growth of the ballooning/interchange instability. We compare the results from our study to previous simulations, and find that the scales mostly match MHD results where the ballooning/interchange instability is initiated by reconnection.

The event seen in our simulation begins with the thinning of the current sheet followed by magnetic reconnection. This results in Earthward flows first on the dusk, and then the dawn side of Earth. The Earthward flow has similar properties to BBFs, and creates additional FACs in the R1 sense. The width of the Earthward flow intrusions is $\sim 3.5\,R_\mathrm{E}$, and they coincide with a decrease in flux tube entropy and an increase in residual $B_z$. The event shows properties of both BBF-related dipolarization and large-scale flows, and the ballooning/interchange instability.

As the magnetospheric solver of the simulation is coupled to an ionospheric solver, we can also study the ionospheric response to the Earthward flows and vorticity. In the ionosphere, we observe the FACs starting to deform to a patchy structure that coincides with the vorticity observed in the transition region. The FACs appear to flow in the R1 sense, and are coupled to the Earthward flow. Our results expand on the previous literature on the connection between the dynamics of fast flows/BBFs in the near Earth tail, and the large scale manifestations of the ballooning/interchange instability.



*Code and data availability.*

Vlasiator is shared under the GPL-2-open-source license at (Pfau-Kempf et al., 2024). The dataset used for this study can be accessed via Suni and Horaites (2024). The Analysator package (Battarbee et al., 2021) was used to perform the analysis for this paper, along with the VisIt (Childs et al., 2012) visualisation tool.

*Video supplement.*

An animation is provided to supplement Figure 2.

*Author contributions.*

VK carried out the analysis and visualisation and wrote the initial draft of the manuscript. VK, MG, MP and EK conceptualised the study and interpreted the results. MA, AW, and ST assisted in the visualisation of the results. LJ, IZ, AW, GC, LP, 375 MA, ST, and KH assisted with the interpretation of the results. UG and YPK developed the simulation used for the study. YPK and JS ran the simulation that was analysed. All authors reviewed the manuscript and gave their comments.

*Competing interests.*

An author is a member of the editorial board of Annales Geophysicae.

*Acknowledgements.* VK, EK, MP and MA acknowledge the Research Council of Finland grant number 352846 (FORESAIL). MA and 380 MP also acknowledge grant number 361901 and the Inno4Scale project via European High-Performance Computing Joint Undertaking (JU) under Grant Agreement No 101118139. The JU receives support from the European Union's Horizon Europe Programme. MG acknowledges funding from the Research Council of Finland (grant 360433-ANAON). YP acknowledges the Research Council of Finland grant 339756 (KIMCHI). GC is supported by the Integration Fellowship of Le Studium Loire Valley Institute for Advanced Studies. The work of JS was made possible by a doctoral researcher position at the Doctoral Programme in Particle Physics and Universe Sciences funded by the 385 University of Helsinki. ST acknowledges the Research Council of Finland grant numbers 336805 and 352846 (FORESAIL), 335554 (ICT-SUNVAC), and 345701 (DAISY). AW acknowledges the Research Council of Finland grant number 347795 (HISSA).

The authors thank the Finnish Computing Competence Infrastructure (FCCI), the Finnish Grid and Cloud Infrastructure (FGCI) and the University of Helsinki IT4SCI team for supporting this project with computational and data storage resources. The authors wish to acknowledge CSC – IT Center for Science, Finland, for computational resources. The simulation presented in this work was run on the 390 LUMI-C supercomputer through the EuroHPC project Magnetosphere-Ionosphere Coupling in Kinetic 6D (MICK, project number EHPC-REG-2022R02-238).





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
