# Peer review of "Mapping transition region flows to the ionosphere in a global hybrid-Vlasov simulation"

_EGUsphere, 2025_

## Author Comment (AC1)

**Response to reviewer 1, author comments given in bold text:**

General comment

The paper presents investigations of the dipole-magnetotail transition region by means of global hybrid-Vlasov simulations of Earth's magnetosphere. The present run of the employed Vlasiator code is merged with an ionospheric solver, and the ionospheric field-aligned currents are related to the magnetospheric vorticity, as a proxy to auroral dynamics. The focus of the paper is on a wave-like density structure that appeared in the transition region after magnetotail reconnection. The wave-like structure is formed by earthward flows with ion vortices on their sides. The authors attribute the wave-like structure to ballooning/interchange activity.

**-The authors thank the referee for the comments and suggestions. We will revise the paper according to the comments and have added several new references as suggested by the reviewer. These additional references will give a more thorough view of previous work into similar topics. The manuscript will be improved by the implementation of these suggestions.**

The simulations clearly reveal a development of a Bz/entropy ridge at about -10 R_E (Figure 4b,f), which is apparently the source of further earthward low-entropy (bubble) intrusions (Figure 4c,g,d,h) due to an interchange process. This is indeed similar to the results of recent global high-resolution (down to ~300km) MHD simulations by Sorathia et al., 2020 (10.1029/2020GL088227). At the same, due to multiple differences (e.g. significantly larger scales and velocities of the present low-entropy intrusions), the present simulation better matches the Rice Convection Model simulations of sawtooth events by Sazykin et al., 2002 (10.1029/2001GL014416), Yang et al., 2008 (10.1029/2008JA013635) and Sun et al., 2021 (10.1029/2021GL094097), where interchange instability operates during storms or substorm, unlike quiet growth phase in simulations of Sorathia et al., 2020 (10.1029/2020GL088227). The RCM simulations show that a wide injection boundary around the geosynchronous orbit may break up into multiple injection channels with the local time separation of about 1–2 h, similarly to present simulations.

**Thank you for these new references, we had not come across the RCM studies previously, but they are very relevant to this study, and we will add them to the manuscript. There are indeed several differences between our study and the Sorathia et al. 2020 study. While the instability appears in some ways to be similar in our work and Sorathia's, the onset mechanism is different. In the Sorathia 2020 paper the**

instability is triggered in the growth phase of a synthetic substorm (prior to any reconnection in the tail), while in our case the instability is closely related to the large-scale onset of reconnection in the magnetotail. In both cases the instability is governed by changes in entropy and Bz, but otherwise the mechanism is different.

Our results are more similar to previous MHD studies(e.g. Lapenta et al., 2011 https://doi.org/10.1029/2011GL047742), and RCM studies (e.g. Yu et al., 2017 https://doi.org/10.1002/2017JA024168 and the new papers suggested by the referee). We will expand on this in the revised manuscript, and clarify the differences between our work and the Sorathia study.

Even more so, the authors attribute the appearance of the interchange-unstable magnetotail configuration (Bz/entropy ridge at -10 R_E) to reconnection and loss of density via plasmoid release, which would be similar to the results of Birn et al., 2011 (10.1029/2010JA016083). This is also a different mechanism, as opposed to the mechanism that is based on flux return to the dayside (Hsieh and Otto, 2015, 10.1002/2014JA020925), which was identified to operate in the run of Sorathia et al., 2020 (10.1029/2020GL088227).

We will clarify the differences and similarities between our results and previous work in the revised manuscript. Our interpretation is that the onset of reconnection, similarly to Birn et al., 2011 results in the lowering of entropy, and eventually a plasmoid release. In our simulation this low entropy region spreads over the magnetotail (as the reconnection spreads over the tail), resulting in the region becoming unstable to the ballooning/interchange instability.  Dayside flux depletion also occurs in our simulation (Tao et al., 2025), contributing to the tail current sheet thinning. The entropy depletion in the tail coincides with the low density, fast Earthward  flow regions caused by reconnection in the tail. Thus we attribute the entropy depletion and the instability to the nightside dynamics.

Thus we have a combination of both BBF-like dynamics, and the ballooning/interchange instability. As we model the ion-kinetic physics of the magnetotail, we can self-consistently capture  these phenomena, and go beyond the MHD and RCM descriptions.

Tao, S., Alho, M., Zaitsev, I., Turc, L., Battarbee, M., Ganse, U., Pfau-Kempf, Y., and Palmroth, M.: Magnetospheric convection in a hybrid-Vlasov simulation, EGUsphere [preprint], https://doi.org/10.5194/egusphere-2025-1340, 2025.

The above major points need to be carefully addressed before publication of the paper. In addition to them I also list below a number of minor suggestions, which may help improve the paper.

Specific comments

A clarifying comment on what leads to reconnection triggering in the Vlasiator would be useful.

**We will comment on this in the manuscript. The reconnection is driven both by numerical diffusion and a tearing-type instability.**

Line 31: reference to Sitnov may not be the best one here, and some auroral paper could be cited instead.

**We will change this reference to Partamies et al., 2015 (https://doi.org/10.1002/2015JA021217)**

Line 32: Additional reference could be added here:

Baumjohann, W., G. Paschmann, and H. Lühr (1990), Characteristics of High-Speed Ion Flows in the Plasma Sheet, J. Geophys. Res., 95, 3801–3809

Line 34: Additional references could be added here:

Baumjohann, W., Hesse, M., Kokubun, S., Mukai, T., Nagai, T., & Petrukovich, A. A. (1999). Substorm dipolarization and recovery. *Journal of Geophysical Research*, 104, 24995–25000.

Baumjohann, W. (2002), Modes of convection in the magnetotail, Phys. Plasmas, 9, 3665–3667, doi:10.1063/1.1499116

Ohtani, S., Singer, H. J., & Mukai, T. (2006). Effects of the fast plasma sheet flow on the geosynchronous magnetic configuration: Geotail and GOES coordinated study. *Journal of Geophysical Research*, 111, A01204. https://doi.org/10.1029/2005JA011383

Merkin, V. G., Panov, E. V., Sorathia, K., & Ukhorskiy, A. Y. (2019). Contribution of bursty bulk flows to the global dipolarization of the magnetotail during an isolated substorm. *Journal of Geophysical Research: Space Physics*, 124, 8647–8668. https://doi.org/10.1029/2019JA026872

Line 35: Additional references could be added here:

Angelopoulos, V., et al. (1996), Multipoint analysis of a bursty bulk flow event on April 11, 1985, J. Geophys. Res., 101, 4967–4989.

Sergeev, V. A., V. Angelopoulos, J. T. Gosling, C. A. Cattell, and C. T. Russell (1996), Detection of localized, plasma-depleted flux tubes or bubbles in the midtail plasma sheet, J. Geophys. Res., 101, 10,817– 10,826, doi:10.1029/96JA00460

Line 37: Additional reference could be added here:

Nakamura, R., Baumjohann, W., Klecker, B., Bogdanova, Y., Balogh, A., R`eme, H., Bosqued, J. M., Dandouras, I., Sauvaud, J. A., Glassmeier, K.-H., Kistler, L., Mouikis, C., Zhang, T. L., Eichelberger, H., and Runov, A. (2002). Motion of the dipolarization front during a flow burst event observed by Cluster. Geophys. Res. Lett., 29:1942

Line 39: Additional reference could be added here:

Shiokawa, K., W. Baumjohann, and G. Haerendel (1997), Braking of highspeed flows in the near-Earth tail, Geophys. Res. Lett., 24, 1179–1182, doi:10.1029/97GL01062.

Line 40: Additional reference could be added here:

Ohtani, S., Y. Miyashita, H. Singer, and T. Mukai (2009), Tailward flows with positive B Z in the near-Earth plasma sheet, J. Geophys. Res., 114, A06218, doi:10.1029/2009JA014159.

Panov, E. V., et al. (2010), Plasma sheet thickness during a bursty bulk flow reversal, J. Geophys. Res., 115, A05213,

doi:10.1029/2009JA014743.

The reference to Panov, E. V., et al. (2010) on Multiple overshoot and rebound of a bursty bulk flow (10.1029/2009GL041971) belongs together with Birn et al., 2011.

Also, the following two references could be placed next to Birn et al., 2011 in this line.

Keika, K., et al. (2009), Observations of plasma vortices in the vicinity of flow-braking: A case study, Ann. Geophys., 27, 3009–3017.

Keiling, A., et al. (2009), Substorm current wedge driven by plasma flow vortices: THEMIS observations, J. Geophys. Res., 114, A00C22,

doi:10.1029/2009JA014114.

Line 47: Additional reference could be added here:

Baumjohann, W., Pellinen, R. J., Opgenoorth, H. J., & Nielsen, E. (1981). Joint two-dimensional observations of ground magnetic and ionospheric electric fields associated with auroral zone currents—Current systems associated with local auroral break-ups. *Planetary and Space Science*, 29, 431–435.

Birn, J., & Hesse, M. (2014). The substorm current wedge: Further insights from MHD simulations. *Journal of Geophysical Research: Space Physics*, 119, 3503–3513. https://doi.org/10.1002/2014JA019863

McPherron, R. L., Nakamura, R., Kokubun, S/, Kamide, Y., Shiokawa, K., Yumoto, K, Mukai, T., Saito, Y., Hayashi, K, Nagai, T., Ables, S., Baker, D. N., Friis-Christensen, E., Fraser, B., Hughes, T., Reeves, G., & Singer, H. (1997). Fields and flows at GEOTAIL during a moderate substorm. *Advances in Space Research*, 20, 923–931.

Palin, L., Opgenoorth, H. J., Ågren, K., Zivkovic, T., Sergeev, V. A., Kubyshkina, M. V., Nikolaev, A., Kauristie, K., Kamp, M., Amm, O., Milan, S. E., Imber, S. M., Facskó, G., Palmroth, M., & Nakamura, R. (2016). Modulation of the substorm current wedge by bursty bulk flows: 8 September 2002—Revisited. *Journal of Geophysical Research: Space Physics*, 121, 4466–4482. https://doi.org/10.1002/2015JA022262

Panov, E. V., Baumjohann, W., Nakamura, R., Weygand, J. M., Giles, B. L., Russell, C. T., et al. (2019). Continent-wide R1/R2 current system and ohmic losses by broad dipolarizationinjection fronts. *Journal of Geophysical Research: Space Physics*, 124, 4064–4082. https://doi.org/10.1029/2019JA026521

Sergeev, V. A., Sauvaud, J.-A., Popescu, D., Kovrazhkin, R. A., Liou, K., Newell, P. T., Brittnacher, M., Parks, G., Nakamura, R., Mukai, T., & Reeves, G. D. (2000). Multiple-spacecraft observation of a narrow transient plasma jet in the Earth's plasma sheet. *Geophysical Research Letters*, 27, 851–854.

Line 81: Additional reference could be added here:

Pritchett, P. L., F. V. Coroniti, and Y. Nishimura (2014), The kinetic ballooning/interchange instability as a source of dipolarization fronts and auroral streamers, *J. Geophys. Res. Space Physics*, 119, 4723–4739, doi:10.1002/2014JA019890.

**We thank the referee for the new references, we will review them and add relevant references at the appropriate lines. The manuscript will benefit from this thorough background information.**

Line 222: Could specific time be indicated after "At the Earthward flows"?

**We will add a time here, it will be good to clarify that.**

Figure 4: A plot with the time evolution of the radial profiles of Bz/PV^gamma could be shown here for the times around t=680 s. This plot would show the growth/formation of the Bz/entropy ridge.

[Figure]

**Indeed such a figure will be useful. We give the figure above, will add it to the manuscript and explain it further in the text. In the figure, we show the radial profiles of entropy and Bz every 50 seconds from 580 s to 880 s. There is a clear decrease in flux tube entropy, and a corresponding increase in Bz between t=580 s and t= 780 s. After this, there is a slight increase in entropy, and a decrease in Bz, which is due to the wavy interchange structure increasing in azimuthal size, so that a higher entropy region moves to Y=0 by 880 seconds (see Fig. 4g and Fig. 4h)**

Figure 6 and associated text: Could the authors explain somewhere how the FAC was obtained?

**The FACs are determined from the curl of B close to the inner boundary of the simulation. We will add this to the manuscript.**

Line 322: Midnight may be more appropriate as mid-tail sounds ambiguous when one considers radial distance instead of azimuthal.

**Midnight is indeed the better word to use here, we will make the change.**

Technical corrections

Line 117: It seems that in is missing between done and six dimensions.

**Yes, 'in' is missing from that sentence, that will be fixed.**

---

## Author Comment (AC2)

**Response to reviewer 2, author comments given in bold text:**

General comment

This paper uses the Vlasiator code, a global hybrid-Vlasov simulation of Earth's magnetosphere with a newly included ionospheric boundary model, to study the formation, evolution, and impact of azimuthally localized fast flows through the magnetotail transition region, defined here to be between ~6-12 $R_E$. The author's show that reconnection first occurs on the dusk, and then dawnside, flanks before extending across the entire magnetotail as seen in the flow reversal between tailward and Earthward flow in Fig. 2f. They show that the region of fast flow that forms symmetrically in the tail at X ~ -8 $R_E$ coincides with low flux tube entropy and increased magnetic field, which becomes unstable to the ballooning interchange instability, driving density and velocity fluctuations with wavelengths ~3.5 $R_E$. Braking of the fast flows causes rebound flows to form and vorticity, which drives FACs into the ionosphere. The authors state that the flows emerge in the simulation after the inclusion of the new ionospheric boundary model, highlighting the importance of magnetosphere-ionosphere coupling.

The authors compare their results to previous works and find that their results are consistent with MHD simulations of low-entropy Earthward flows driven by reconnection (e.g. Birn & Hesse 2013) rather than those where the instability is driven by magnetic flux evacuation to the dayside during substorm growth phase (Sorathia 2020). They postulate that, in the current simulation, these features, both in the magnetosphere and their auroral counterparts, are dominated by larger scales rather than kinetic-scale processes. While the comparison to previous works is extremely helpful to put the results into context, a clearer distinction on the new insights provided by this work would help set this paper apart from the others. Additional comments are below.

**We thank the referee for the insightful comments that will improve the manuscript quality, and will make changes accordingly.**

**We will elaborate further on the new insights available through this work in the manuscript. Our model captures the dynamics of BBF-like flows, tail-wide entropy depletions, current sheet thinning and reconnection, and the mapping of these phenomena to the ionosphere. We thus model the several different types of interconnected phenomena that have been seen in other simulations, combining the effects in a self-consistent manner. This is the first time that a similar event has been seen in a hybrid-Vlasov simulation, where the ion dynamics are captured. Our**

**simulation offers a perspective on the development of several BBF flow channels (in the presence of ion-kinetic physics) that form from a single wide reconnection region in the close tail.**

**This is a possible explanation for the "wedgelet" phenomenon, where several pairs of FACs are observed in the ionosphere. In our simulation we see the transition from a the classic R1/R2 ionospheric current pattern (before the large inflow region splits into several flow channels) to a "wedgelet" type current distribution, associated with multiple magnetospheric flow channels.**

Specific Comments

-Line 173: could reference Figure 3c and 4a-d when referring to the Bz enhancement as scale makes it difficult to identify in Figure2i-l.

**We can add a reference here to Figure 3c as well.**

-The reconnection starts very close to Earth despite those events being relatively rare (Beyenne & Angelopoulos 2024). Whether this is the first time reconnection occurs in the tail would be helpful to note. The initial state seems to be a dipole field and constant IMF (line 218). The reconnection shown occurs about 10 minutes into the simulation so it is unclear if this the first time reconnection is occurring in the tail as the magnetotail forms or if magnetosphere has been sufficiently preconditioned and is not significantly affected by the wave of IMF as it passes the magnetosphere for the first time.

Beyene, F., & Angelopoulos, V. (2024). Storm-time very-near-earth magnetotail reconnection: A statistical perspective. Journal of Geophysical Research: Space Physics, 129, e2024JA032434. https://doi.org/10.1029/2024JA032434

**This is the first onset of reconnection we see in the simulation. We have a comment on this in the Discussion (line 327), but the point will be elaborated to further consider the effects of the early stage of the simulation. The simulation results could be compared to a magnetospheric situation where the solar wind speed rapidly rises after a period of slow wind. The FAC system has already been initialized in the traditional R1/R2 sense, and so we can study the mapping between the ionospheric and magnetospheric domains. We thank the referee for suggesting this new reference, it is very relevant to our study and we will add it to the manuscript. It is also to be noted here that the ion-kinetic description of the entire magnetosphere shows**

that reconnection is ubiquitous in the magnetotail, occurring in many different regions simultaneously (see e.g., Alho et al, Palmroth et al 2023).

Palmroth, M., Pulkkinen, T.I., Ganse, U. *et al.* Magnetotail plasma eruptions driven by magnetic reconnection and kinetic instabilities. *Nat. Geosci.* 16, 570–576 (2023). https://doi.org/10.1038/s41561-023-01206-2

Alho, M., Cozzani, G., Zaitsev, I., Kebede, F. T., Ganse, U., Battarbee, M., Bussov, M., Dubart, M., Hoilijoki, S., Kotipalo, L., Papadakis, K., Pfau-Kempf, Y., Suni, J., Tarvus, V., Workayehu, A., Zhou, H., and Palmroth, M.: Finding reconnection lines and flux rope axes via local coordinates in global ion-kinetic magnetospheric simulations, Ann. Geophys., 42, 145–161, https://doi.org/10.5194/angeo-42-145-2024, 2024

In the movie within the supplemental information, the region where Vx > 400 km/s appears to first extend in MLT across the tail before driving earthward flows. Is this region being continuously driven by reconnection? Showing radial profiles of the $B_z$ and flux tube entropy in the tail as a function of time would be helpful to show why the region becomes unstable to the ballooning instability later in the simulation and then dissipates. 2D simulations (Zhu et al. 2004) have shown that the plasma beta can affect the growth rate of the ballooning mode for sufficiently thin current sheets. The evolving state of the tail, therefore, might be affecting when the density fluctuations occur.

Zhu, P., A. Bhattacharjee, and Z. W. Ma (2004), Finite ky ballooning instability in the near-Earth magnetotail, J. Geophys. Res., 109, A11211, doi:10.1029/2004JA010505.

We thank the referee for this insight into the evolution of the density fluctuations. A figure showing the Bz and flux tube entropy profiles will be added to the manuscript, and is also shown below. In the figure, we show the radial profiles of entropy and Bz every 50 seconds from 580 s to 880 s. There is a clear decrease in flux tube entropy, and a corresponding increase in Bz between t=580 s and t= 780 s. After this, there is a slight increase in entropy, and a decrease in Bz, which is due to the wavy interchange structure increasing in azimuthal size, so that a higher entropy region moves to Y=0 by 880 seconds (see Fig. 4g and Fig. 4h)

We observe a reconnection X-line throughout the studied interval, and we see dipolarization in the Bz component as the event progresses. We will add more discussion on this into the manuscript.

[Figure]

-In section 4 of the discussion, clarification on what is setting the wavelength of the density fluctuations and fast flows would be helpful to determine if it is spatially localized reconnection or the ballooning interchange instability itself. If reconnection sets the wavelength, then clarification on how it is generating that wave-like structure and whether it is bursty, or continuous would help shed light on why the flows have the widths that they do.

**The reconnection appears to be continuous, spreading over the tail from dawn and dusk towards midnight. This results in a cross-tail X-line forming across the magnetotail, and a corresponding decrease in flux tube entropy. After this, we see signs of the ballooning/interchange instability. The wavelength is not related to bursty reconnection that would form BBF-like structures, but rather the structures form on the Earthward side of the cross-tail X-line. The eventual wavelength of the density fluctuation is the same as the original flow channels that intrude into the transition region.**

Technical Corrections

line 183: remove "along with an additional" from "an additional along with an additional upward current"

**Thank you, this will be removed.**